# Prolonging the Half-Life of Histone Deacetylase Inhibitor Belinostat via 50 nm Scale Liposomal Subcutaneous Delivery System for Peripheral T-Cell Lymphoma

**DOI:** 10.3390/cancers12092558

**Published:** 2020-09-08

**Authors:** Meng-Hsuan Cheng, Jun-Yi Weng, Chih-Hung Chuang, Wei-Ting Liao, Yu-Fong Lai, Jia-Yu Liu, Yi-Ping Fang

**Affiliations:** 1Department of Internal Medicine, Division of Pulmonary and Critical Care Medicine, Kaohsiung Medical University Hospital, Kaohsiung 80708, Taiwan; 880298@kmuh.org.tw; 2School of Medicine, College of Medicine, Kaohsiung Medical University, Kaohsiung 80708, Taiwan; 3Department of Respiratory Therapy, College of Medicine, Kaohsiung Medical University, Kaohsiung 80708, Taiwan; 4School of Pharmacy, College of Pharmacy, Kaohsiung Medical University, Kaohsiung 80708, Taiwan; alex_weng7@hotmail.com (J.-Y.W.); yufong74@gmail.com (Y.-F.L.); u109830006@kmu.edu.tw (J.-Y.L.); 5Department of Medical Laboratory Science and Biotechnology, College of Health Sciences, Kaohsiung Medical University, Kaohsiung 80708, Taiwan; a4132600@kmu.edu.tw; 6Drug Development and Value Creation Center, Kaohsiung Medical University, Kaohsiung 80708, Taiwan; 7Department of Biotechnology, College of Life Science, Kaohsiung Medical University, Kaohsiung 80708, Taiwan; wtliao@kmu.edu.tw; 8Regeneration Medicine and Cell Therapy Research Center, College of Medicine, Kaohsiung Medical University, Kaohsiung 80708, Taiwan; 9Department of Medical Research, Kaohsiung Medical University Hospital, Kaohsiung Medical University, Kaohsiung 80708, Taiwan

**Keywords:** belinostat, peripheral T-cell lymphoma, liposome, drug delivery system

## Abstract

**Simple Summary:**

Belinostat is the novel histone deacetylase inhibitors (HDACis) for treatment for peripheral T-cell lymphoma (PTCL). However, the half-life of belinostat is only 1.1 h. The aim of the study was to improve the half-life and it’s in vivo circulation behavior by using liposome encapsulation technology. The 50 nm scale liposomes were prepared, which showed the sustained release behavior, decrease the burst effect and improving the drug’s toxicity and had similar power for HuT-78 cells. Moreover, we proposed that phospholipid types are crucial factors for size forming and in vivo circulation behavior. We found that DOPC phospholipid material increased the half-life of belinostat, decreased clearance and presented a higher area under curve exposure. Due to the lymphatic delivery complexation, the localized at the lymphatic organs study is necessary to evaluate in the near future.

**Abstract:**

Lymph node metastasis is an aggressive condition characterized by poor treatment outcomes and low overall survival. Belinostat is a novel histone deacetylase (HDAC) inhibitor approved by the Food and Drug Administration (FDA) for the treatment of relapsed peripheral T-cell lymphoma (PTCL). However, the major problem is that belinostat has a short half-life of 1.1 h. In this study, we successfully prepared 50 nm liposomal colloids, which showed a controlled release pattern and excellent pharmacokinetics. The results showed that the particle size of liposomes consisting of dioleoylphosphatidylcholine (DOPC) was larger than that of those consisting of dioleoylglycerophosphoserine (DOPS). In terms of release kinetics of belinostat, the free drug was rapidly released and showed lower area under curve (AUC) exposure for in vivo pharmacokinetics. When liposomal formulations were employed, the release pattern was fitted with Hixson–Crowell models and showed sustained release of belinostat. Moreover, HuT-78 cells were able to take up all the liposomes in a concentration-dependent manner. The safety assessment confirmed hemocompatibility, and the platelet count was increased. Furthermore, the liposomes consisting of DOPC or DOPS had different behavior patterns, and their delivery to lymphatic regions should be thoroughly investigated in the future.

## 1. Introduction

Lymph node metastasis is an aggressive condition characterized by poor treatment outcomes and low overall survival. Peripheral T-cell lymphoma (PTCL) is a highly heterogeneous malignancy accounting for 10–15% of all non-Hodgkin’s lymphomas in the Western world, and its incidence is higher in East Asia [1]. In contrast to solid tumors, PTCL is spread widely throughout the bone marrow and blood system [2]. Surgical resection is impossible, and radiotherapy and chemotherapy rarely result in positive treatment outcomes [3]. Based on the current research, the treatment of lymphatic metastatic tumors is an unmet medical need.

The standard first-line chemotherapy treatment for PTCL uses a combination of cyclophosphamide, doxorubicin, vincristine and prednisone (CHOP). However, the results have a poor prognosis, with median overall survival (OS) of 6.5 months due to rapid relapse, even when using CHOP plus etoposide (CHOPE) [4]. Several therapeutic approaches have been developed, including histone deacetylase inhibitors (HDACis), immunoconjugates, monoclonal antibodies (mAbs), immunomodulatory agents, nucleoside analogs, proteasome inhibitors and kinase inhibitors [5]. Among the therapeutic approaches, belinostat (PXD101) was granted orphan drug designation, which was approved by the FDA in July 2014, being given fast-track designation as a treatment for relapsed or refractory PTCL [6].

Belinostat has a sulfonamide–hydroxamide structure which binds with the zinc finger on HDAC, causing the accumulation of acetylated histones and other proteins, and increasing the expression of tumor-suppressor genes [7]. Belinostat (C_15_H_14_N_2_O_4_S) has a small molecular structure (318.35 g/mol) with moderate solubility (0.08 mg/mL at pH 7.4), properties that make it a suitable drug candidate. However, the half-life of belinostat is only 1.1 h [8]. It is imperative to develop a method to prolong the half-life of belinostat. Panobinostat, vorinostat, and chidamide are existing oral dosage forms, but their low bioavailability of about 21–43% is a major limitation [9].

The predominant challenge of lymphatic targeting is the complex network of the lymphatic system. The lymphatic system is a one-way, open-ended transit network that consists of lymphatic vessels, lymph nodes, spleen, thymus, Peyer’s patches and tonsils, and which also plays a crucial role in immune surveillance and response function [10]. To improve the limitations of lymphatic system transport, various carrier systems have been broadly investigated. Most researchers agree that the particle size is significant [11].

The penetration of particles is generally inversely proportional to their size. Smaller particles (<100 nm) more easily penetrate the microvasculature of lymphatic metastatic tumors, thereby enhancing the antitumor efficiency [12]. Furthermore, drugs can be passively targeted to lymphoma through the high accumulation of nanometer carriers during lymphatic metastasis [13], but this is relative to the carrier origin type [14].

However, a few previous studies have used nanotechniques to deliver HDAC inhibitors in cancer applications. Martin et al. demonstrated that belinostat-loaded poly(lactide-co-glycolide) (PLGA) decorating poly(guanidinium oxanorbornene) and DSPE-PEG can enhance cytotoxicity and work against bladder cancer [15]. A study by Urbinati et al. involved the encapsulation of belinostat using a liposome technique for breast cancer treatment [16]. It was proposed that phospholipid is a crucial factor in the belinostat-loaded liposomal delivery system. The liposomal delivery system is the most common and promising nanocarrier delivery system for targeted drug delivery. Liposomes have a number of demonstrated benefits including enhanced cellular uptake, improved biodistribution behavior, increased circulation half-life and minimized systemic toxicity [17]. Moreover, they exhibit low toxicity and immunogenicity and high biodegradability characteristics, which allow them to be widely used in biological systems [18]. Importantly, past research has pointed out that liposome carriers have potentially accumulated at the lymph nodes [19].

The purpose of this study was to optimize the belinostat-loaded liposome delivery system and characterize its physical properties and in vitro release pattern to improve its circulation behavior. Human T-cell lymphoma cellular systems (HuT-78) were used for in vitro cellular viability testing and the uptake assay. In vivo animal pharmacokinetics tests were performed to clarify the pharmacokinetics behavior.

## 2. Results

### 2.1. Characterization of Belinostat-Loaded Liposome Delivery System

Figure 1 shows the schematic diagram of various liposomes incorporating dioleoylphosphatidylcholine (DOPC), dioleoylglycerophosphoserine (DOPS) and DOPC/DOPS groups. Table 1 shows the characteristics of various types of belinostat-loaded liposomes including DOPC, DOPS and mixed DOPC/DOPS formulations. The particle size of liposomes in the DOPC group was 66.65 ± 3.15 nm, which was significantly larger than those of the DOPS or mixed DOPC/DOPS groups (*p* < 0.05). When incorporating the DOPS or mixed DOPC/DOPS groups, the particle diameters were approximately 50 nm. In addition, the zeta potential of liposomes with the DOPC group was −10.18 ± 1.95 mV, which was significantly higher than that of other groups. In the DOPS and mixed DOPC/DOPS groups, the zeta potential showed stronger negative charges at −64.27 ± 4.01 and −70.00 ± 5.79 mV, respectively. The entrapment efficiency of the DOPC group was significantly higher than that of the DOPS or mixed DOPC/DOPS groups (*p* < 0.05). Observation of DOPC showed some cloudy colloids, while the DOPS and mixed DOPC/DOPS groups were clear and transparent (Figure 2a). Belinostat-loaded liposomes showed nanometer-sized sphericity, as seen in transmission electron microscopy (TEM) images (Figure 2b).

### 2.2. In Vitro Release Kinetics and Mechanisms

The release profile is shown in Figure 3a. The release kinetics of the belinostat-free drug group was completely exhausted within 4 h, as well, 80% of the initial burst release occurred within 2 h. Various belinostat-loaded liposomal groups all presented a biphasic and sustained release pattern and showed linearity from 0 to 4 h, followed by saturation at 24 h. There were no significant differences among the belinostat-loaded DOPC, DOPS, and mixed DOPC/DOPS liposome groups (*p* > 0.05). The cumulative release percentage data were plotted and fitted based on five different mathematical models: zero-order, first-order, Higuchi, Korsmeyer–Peppas and Hixson–Crowell, as shown in Figure 3b. The free belinostat control group was fitted with the Higuchi model (R^2^ = 0.979). All belinostat-loaded liposomal groups fit the Hixson–Crowell model best (R^2^ > 0.992). Table 2 illustrates the rapid release mechanism of belinostat from true solution, and the belinostat-loaded liposomal carrier exhibited a sustained release pattern.

### 2.3. FTIR Characterization

Figure 4 shows the FTIR spectra of belinostat free drug, empty liposomal formulations and belinostat-loaded liposomal formulations. Belinostat exhibited main infrared peaks following O–H stretching vibrations in the hydroxyacrylamide at 3235 cm^−1^, N–N stretching in the secondary amine at 3215 cm^−1^, C–H stretching in the aromatic at 3040 cm^−1^ and S=O stretching in the sulfonamide at 1153 and 1340 cm^−1^. All liposome groups showed symmetric and antisymmetric stretching vibrations of the CH_2_ in the acyl chain at 2850 and 2920 cm^−1^. The headgroup of DOPC showed stretching at N^+^(CH_3_) at 970 cm^−1^. The fingerprint of belinostat did not appear in the spectra of any of the liposome formulations. The spectra of each formulation were similar to those of the respective empty liposome groups.

### 2.4. In Vitro Cytotoxicity and Cellular Uptake Study

The cytotoxicity and IC_50_ value of belinostat free drug and belinostat-loaded liposomal formulations on HuT-78 cell lines are shown in Figure 5a. The results showed that belinostat free drug inhibited HuT-78 proliferation in a dose-dependent manner, and a sigmoid curve was obtained at a range of 0.1 to 10 μM. After treatment with belinostat-loaded liposomal formulations, HuT-78 cell viability was 30%, and the IC_50_ value was 1.73–1.95 μg/mL. In both the cytotoxicity and IC_50_ value, there were no significant differences (*p* > 0.05). Figure 5b shows the cellular uptake in HuT-78. All the liposomal groups showed dose-dependent cellular uptake in HuT-78, but there were no significant differences between the different phospholipid materials.

### 2.5. In Vivo Pharmacokinetics and Biodistribution

Figure 6a shows the mean plasma concentration vs. time for belinostat free drug and belinostat-loaded liposomal formulations. The pharmacokinetic parameters were calculated according to the equation in Figure 6b and are summarized in Table 3. The data showed lower maximum plasma concentration (C_max_) in the belinostat free drug group. Three evaluated liposomal groups were increased about 1.5–2-folds in C_max_ compared to the belinostat free drug group. Moreover, DOPC liposomes showed a different pattern, which presented sustained release behavior despite the lower C_max_. The DOPS or DOPC/DOPS mixture groups showed higher C_max_ (*p* < 0.05), but this decreased rapidly after T_max_. AUC and half-life values of the belinostat-loaded DOPC liposome group were significantly higher than that of the belinostat control group and other liposomal groups. Moreover, the clearance of DOPC liposomes was lower than other groups, but there were no significant differences (*p* > 0.05). Both the DOPC and DOPS liposomes showed higher accumulative amounts in spleen tissue by about 1.5–2-fold, and in lungs by about 1.3–1.4-fold when compared to the belinostat free drug group (Figure 6c).

### 2.6. Toxicity Assessment

Hemolysis activity was evaluated to ascertain the biocompatibility between red blood cells and liposome carriers, as shown in Figure 7a. There was no hemoglobin measured in the tube in all the belinostat-loaded samples in the liposomal group whether by UV detection or observation. Moreover, the hemoglobin level was the same as in the negative control. The results showed almost no hemolysis activity in any liposomal groups. Figure 7b shows that the viabilities of peripheral blood mononuclear cells (PBMCs) were higher relative to belinostat concentration. When treated with 0.1 μM of belinostat, the PBMCs were safer. When treated with 10 μM of belinostat, the viability of PBMCs remained only 20%. Moreover, we evaluated complete blood counts and liver function markers of toxicity during subcutaneous administration in vivo in mice, as shown in Table 4 The data showed that the free belinostat control group had values similar to the reference values, except for platelet count and hematocrit, which were lower than reference values. All the complete blood count (CBC) data of liposome groups were similar to reference values. Moreover, liposomes in the DOPC group improved the platelet count compared to the reference value. In both liposomal groups, the liver function markers alanine aminotransferase (ALT) and aspartate aminotransferase (AST) were within the reference values.

## 3. Discussion

Belinostat is an HDAC inhibitor approved by the FDA for the treatment of relapsed peripheral T-cell lymphoma. However, its short half-life is a major drawback. To achieve higher circulation behavior in vivo, we developed a liposomal delivery system. All the prepared liposomes were stable and maintained at around a 50 nm scale for 2 months. In subcutaneous injection with liposomal formulations, the burst release pattern was decreased along with the sustained release of belinostat. The pharmacokinetics study demonstrated that the investigated liposome delivery system improved the circulation time of belinostat in plasma.

Phospholipid diversity is attributed to the combination of head groups with hydrocarbon chains that vary in fatty acid length, double bond number and position. DOPS and DOPC are both unsaturated phospholipids with dioleoyl fatty acid chains (18:1, △9-cis), differing only in their headgroups. The transition temperatures of DOPC and DOPS were −22 °C and −10 °C, respectively. The theory of packing parameter states *P = V/a_o_ lc*, where *P* is the critical packing parameter (shape factor), *V* is the specific volume occupied by the tails, *a_o_* is the area per lipid molecule in the dividing surface and *lc* is the effective maximum length of the tails’ region. *P* characterization is important for predicting structural geometry; *P*~1 denotes the lamellar phase, while *P* > 1 indicates the cone phase. In the fluid state, where lipids rotate freely, a critical packing parameter of about 1 in DOPC resulted in a lamellar phase being formed. A critical packing parameter other than 1 in DOPS led to a hexagonal II(cone) phase being formed [20]. Previous studies showed that the headgroup areas (nm^2^) of DOPC and DOPS were 0.725 and 0.65, respectively [21,22]. In the present study, the particle size of the DOPS group was significantly smaller than that of the DOPC group. The main affecting factor was assumed to be the headgroup, as the geometric packing property of DOPS with smaller headgroup areas formed larger vesicles and shapes with less curvature. The encapsulation efficiency of DOPS was significantly lower than that of the DOPC group. This phenomenon can be explained by the charge properties of DOPS, which are acidic and negatively charged, the effects of the electrostatic head group interaction, increased packing defects and loose packing. In addition, during the preparation process, probe sonication was a generated thermal effect. In the thermal process of probe sonication, the DOPC group appeared cloudy due to lower Tc (−22 °C). On the other hand, the DOPS or DOPC/DOPS mixture groups were apparently clear, with the Tc (−10 °C) slightly higher than that of the DOPC. Therefore, the appearance was markedly different. FTIR is a useful tool for evaluation of the surface chemistry of particles after modification and changes. In the present study, the major functional groups of belinostat are N–H and S=O, and the presence of these functional groups had no influence at the same wavenumber of liposomal formulations. The results suggested that belinostat molecules are encapsulated at their phospholipid bilayers.

The belinostat free drug showed a burst release pattern in the in vitro dialysis platform. However, the in vivo pharmacokinetics behavior displayed lower plasma concentrations of belinostat. Therefore, the burst-released belinostat did not make it out of the range of the therapeutic window. However, the lower plasma concentration should be improved. In terms of the mechanism study, several plausible mathematical models were calculated. All belinostat-loaded liposomal formulations were best fitted with the Hixson–Crowell model (R^2^ > 0.99). The Hixson–Crowell model mostly describes release from dosage systems where there is a change in the surface area and diameter of particles or tablets, which is proportional to the cube root of their volume [23]. A previous study used phosphotidylcholine and phosphotidylserine, which had the same release pattern [24]. The mechanism of slow release can be interpreted in terms of a change in surface area during the dissolution process, which has a significant effect on drug release. Moreover, this is a possible reason for the strong affinity between belinostat and the phospholipid. In addition, all the liposomal formulations displayed the same release model due to the similar liposomal particle volume. Hence, when proportional to the cube root of the volume, a similar trend was observed. On the other hand, the sustained release behavior was also attributed to the incorporation of cholesterol. Research has reported slow drug-release behavior to be characteristic of a rigid liquid-ordered liposome. When a high amount of cholesterol is present in the liposome, the lipid bilayer should be in a rigid liquid-ordered phase and thus enable a prolonged drug-release pattern [25].

According to in vitro cellular evaluation of HuT-78, we found that the belinostat free group and liposomal formulations showed a similar IC_50_. However, the cytotoxicities of the belinostat free drug and belinostat-loaded liposomal formulations were sensitive to drug concentration. The sigmoidal curves were obviously in the highly sensitive range at 1 to 5 mM of belinostat. Rhodamine, a hydrophobic fluorescent molecular probe, is an excellent tool for determining liposome-particle retention in cells for bioassays [26]. The results of flow cytometry confirmed that HuT-78 can uptake belinostat-loaded liposomes in a concentration-dependent manner.

Pharmacokinetic parameters demonstrated that with subcutaneous administrated belinostat-loaded liposome, the DOPC group slightly prolonged the half-life in the bloodstream, had large AUC exposure and sustained release behaviour in plasma concentration vs. time profile. We assumed this to be relevant to the lower clearance of the DOPC liposome. In terms of the DOPS group, the clearance was similar to the free control group, and belinostat-loaded liposome may rapidly decrease in the blood stream. Combined with biodistribution data, we found that liposomes with the DOPS group were highly localized at the spleens and lungs. These results indicated that the DOPC material helped to increase vesicle circulation time, but the ability to escape the mononuclear phagocytic system (MPS) uptake was limited. In the treatment of such aggressive lymphoma diseases, the lymphatic target is of significant importance. Previous literature has reported that phosphotidylserine can be recognized by mononuclear phagocytes [27]. The reorganization system helps to identify negatively charged phospholipids, especially phosphotidylserine. This was supported by our finding that higher accumulative amounts of both the liposome types were localized in the spleen and lungs in comparison with the belinostat control group.

The toxicity of liposomes was evaluated to ensure minimal destruction of red blood cells. We detected hemoglobin at UV wavelengths and the results showed that no hemolysis occurred in any liposomal formulations. This demonstrated that there was no interaction between the liposomes and blood constituents during administration of liposomes, whether DOPC or DOPS. Past research has pointed out that short chain phosphatidylcholines can be spontaneously transferred between the liposomes and erythrocyte membranes. Moreover, the order of hemolysis rate was increased by the fatty acyl chain in shorter and incorporated amounts [28,29]. In our case, the long fatty acyl chain indeed averted hemolytic activity.

On the other hand, the cytotoxicity of PBMCs showed sensitivity in a dose-dependent manner in the range 0.1 to 10 μM. Our preliminary data showed that individual differences occurred. PBMCs are a diverse mixture of highly specialized immune cells that play key roles in keeping our bodies healthy [30]. Therefore, individual differences frequently occurred. PBMCs are critical tools for determining the dosage limit of new drug compounds and nanoparticles. They provide the toxicity potential of new compounds on the human immune system [31]. Based on the results, the treatment dose is a key factor influencing PBMCs. In this case, the liposomal formulation modification did not change the toxicity of PBMCs.

In order to understand the safety profile during the administration period, complete blood count (CBC) and liver function were monitored. Low red blood cell counts occurred in patients more than 30% of the time when using intravenous (IV) infusion over 30 min [32]. In the case of the in vivo study, red blood cell (RBC) count of the belinostat control group demonstrated values similar to the reference value of subcutaneous administration. Liposome groups followed the same trend. These phenomena indicated that the safety profile can be changeable when using the subcutaneous administration route. Hematocrit was slightly decreased when compared to the reference value in the subcutaneously administered belinostat control group. In contrast to the liposome system, the hematocrit was significantly increased (*p* < 0.05) and similar to the reference value. Overall, the belinostat-loaded liposome system improved the anemia side effect. Previous literature also showed that low platelet count occurred in about 10–29% of patients [26]. The safety profile showed that the platelet count was significantly lower than both liposome systems (*p* < 0.05). This indicated that a low platelet counts still occurred after subcutaneous administration of belinostat. However, the platelet counts were similar to reference values in both liposome systems.

## 4. Methods

### 4.1. Materials

Belinostat was purchased from Active Biochem (HK). The 1,2-dioleoyl-sn-glycero-3-phosphocholine (DOPC) and 1,2-dioleoyl-sn-glycero-3-phospho-L-serine (DOPS) were obtained from Avanti Polar Lipids (Alabaster, AL, USA). Carbamazepine, cholesterol, rhodamine B, and uranyl acetate were purchased from Sigma-Aldrich (St. Louis, MO, USA). Sodium phosphate dibasic (Na_2_HPO_4_) and citric acid were obtained from Merck (Darmstadt, Germany). Ethanol, chloroform, and acetonitrile of analytical reagent grade were obtained from Mallinckrodt (Staines-upon-Thames, UK).

### 4.2. Preparation of Belinostat-loaded Liposomes

Belinostat-loaded liposomes were prepared using the thin-film method. The formulation was composed of DOPC, DOPS as phospholipid material, CHEMS, and DSPE-PEG2000 folate as the lipid phase which was dissolved in 99.5% ethanol. The mixture was evaporated using a rotary evaporator at 55 °C, and residual solvents were removed using nitrogen for 10 min. The film was hydrated with chloroform or absolute ethanol for DOPC and DOPS phospholipids, respectively. The hydrated liposome coarse colloids were dispersed using a probe sonicator (Branson Digital Sonifier^®^ Model 250; Branson Ultrasonics Corp., Danbury, CT, USA) for 5 min at 30 W. The liposomes were then extruded stepwise through a polycarbonate membrane with pore size of 70 to 100 nm 15 to 20 times in a mini-extruder (Avanti Polar Lipids, cat. no. 610000, Alabaster, AL, USA) and settled at 65 °C.

### 4.3. Characterization of Particle Diameter and Zeta Potential

The particle diameter (d) and zeta potential (ζ) of liposomes were measured via dynamic light scattering (DLS) (ELSZ-2000, Otsuka Electronic, Hirakata, Japan). The polydispersity index (PdI) was used to measure the size distribution. All vesicles were diluted 200-fold with double-distilled water to achieve the count rate for the measurements. The determination was repeated three times for each of the three samples.

### 4.4. Chromatographic Conditions and Validation of Belinostat In Vitro

Belinostat was analyzed by high-performance liquid chromatography (HPLC). The HPLC system consisted of an L-7100 pump, L-7200 autosampler, L-7455 diode array detector at 265 nm (Hitachi, Tokyo, Japan) and a Purospher Star RP-18 endcapped column (250 × 4.6 mm, internal diameters 5 μm, Merck). The mobile phase was a mixture of 0.1% phosphoric acid and acetonitrile by gradient elution, and the flow rate was 1 mL/min. The column oven was set at 30 °C. Limits of detection and quantitation of belinostat were determined by dissolving belinostat at decreasing concentrations in distilled deionized water until the signal/noise ratios were 3 and 10, respectively. The linearity of the standard curves and intraday and interday precision and accuracy were established.

### 4.5. Measurement of Encapsulation Efficiency

The separation of the encapsulated (free) portion and unencapsulated (free) portion was achieved using an ultracentrifugation method. Fresh liposomes were centrifuged at 771,000× *g* for 1 h at 4 °C in a Hitachi CS150 GXL ultracentrifuge. The supernatant was collected to quantify the unencapsulated drug. The original fresh liposomes were dissolved in chloroform and methanol (1:1) and centrifuged at 12,000 rpm for 10 min at 4 °C to measure the total drug content. The encapsulation efficiency of belinostat-loaded liposomes was calculated using the following equation:(1)$$\mathrm{Encapsulation}\;\mathrm{Efficiency}\;\left(\%\right)=\;\frac{Concentration_{total}-Concentration_{free}}{Concentration_{total}},$$
where Concentration_total_ is the amount of drug placed in the formulation, and Concentration_free_ is the supernatant collected after centrifugation.

### 4.6. TEM

Morphology was observed by transmission electron microscopy (TEM). Copper grids were used in plasma pretreatment to increase hydrophilicity. The liposome suspension was dropped onto a Formvar/carbon film on a 200-mesh copper grid (FCF-200-Cu; Electron Microscopy Sciences, Hatfield, PA, USA) and stained with 1% uranyl acetate for 1 min. After the negative-staining process, the sample was washed with distilled deionized water. The water was vacuumed out overnight and the sample was then observed using TEM (JEM-1400; JEOL, Tokyo, Japan).

### 4.7. Stability Test

A series of liposomal formulations were divided into separate vials and stored at 4 °C. The particle size, polydispersity and zeta potential of all the formulations were determined after 7, 14, 21 and 28 days.

### 4.8. Fourier-transform Infrared Spectroscopy (FTIR)

The samples were diluted with approximately 1% KBr mixing powder and pressed into 16 mm disks at 15 t for analysis using a Fourier-transform infrared spectrometer (Vertex 70 v, Bruker, Billerica, USA). The wavenumbers ranged from 4000 to 380 cm^−1^ at a resolution of 1 cm^−1^.

### 4.9. Drug Release Profiles

An in vitro release study was performed using a dialysis method. Dialysis bags with a molecular-weight cutoff of 6000–8000 Da (pore size 1.8 nm; Orange Scientific, Braine-l’Alleud, Belgium) were soaked with distilled deionized water for 12 h before the experiment. Dialysis bags containing SN38-loaded liposomes, or SN38 solution, were subsequently incubated in flasks containing 15 mL medium (phosphate buffered saline/ethanol 1:1 *v*/*v*) in a 37 °C water bath. At appropriate intervals, 500 μL aliquots of the medium were withdrawn for analysis and immediately replaced with an equal volume of fresh medium. Release kinetics were investigated using different models, including zero-order, first-order and Higuchi kinetics. Correlation coefficients and each release equation were calculated from the respective plots.

### 4.10. Cell Cytotoxicity

The cutaneous T-cell lymphoma cell line (HuT-78) was purchased from the Bioresource Collection and Research Center (BCRC, Hsinchu, Taiwan). HuT-78 cells were incubated in 79% Iscove’s modified Dulbecco’s medium (IMDM) with 20% FBS and 1% liquid penicillin–streptomycin and kept in a humidified atmosphere containing 5% CO_2_ at 37 °C. HuT-78 cells were seeded on 96 well culture plates at a cell density of 1.5 × 10^5^ cells per well. After overnight incubation, the medium was replaced with fresh medium containing belinostat free drug and belinostat-loaded liposomal formulations at final concentrations of 0.05, 0.1, 0.5, 1, 5 and 10 μM for 72 h. Cells were washed with PBS and reacted with 10 μL Cell Counting Kit-8 (CCK-8, Dojindo, Kumamoto, Japan) for 2 h. The absorbance was read at 450 nm by an enzyme-linked immunosorbent-assay reader (BioTek Epoch; Thermo Fisher Scientific, Waltham, MA, USA) to measure the quantity of surviving cells. IC_50_ was calculated using a four-parameter logistic function standard curve analysis of the dose response. The cell cytotoxicity was calculated using the following equation:(2)$$\mathrm{Cell}\;\mathrm{viability}\;\%=\;\frac{OD_{sample}-OD_{blank}}{OD_{control}-OD_{blank}}\times 100\;\%$$
where *OD* denotes the optical density, *OD**_control_* represents 100% survival, and *OD**_blank_* represents no cells.

### 4.11. Flow Cytometry

Cellular uptake of belinostat-loaded liposomes into the HuT-78 cells was evaluated by flow cytometry (Cytomics FC500, Beckman Coulter, Brea, CA, USA). Rhodamine B was incorporated in liposomes as a fluorescence indicator. HuT-78 cells were seeded into six-well plates at 1.5 × 10^5^ cells per well and incubated overnight. The medium was removed and replaced with fresh medium containing belinostat-loaded liposomes at final concentrations of 10 μM for 4 h. The mixture was centrifuged at 1000 rpm for 5 min and the supernatant was removed. Cell pellets were then washed twice with PBS, and cells were suspended in PBS and analyzed by flow cytometry. The fluorescence signal was read at 488/575 nm (excitation/emission) by the FACScan system.

### 4.12. In Vivo Pharmacokinetics Study

#### 4.12.1. Animal

The in vivo pharmacokinetic study was performed on healthy male BALB/c mice (weight: 16–20 g) obtained from the Laboratory Animal Center of the National Science Council, Taipei, Taiwan. All animal experiments were conducted in accordance with institutional guidelines and approved by the Animal Care and Use Committee of Kaohsiung Medical University, Kaohsiung, Taiwan (KMUIACUC-107146). All animals were starved overnight prior to the experiments.

#### 4.12.2. Administration of Belinostat-Loaded Liposomes

BALB/c mice were anesthetized by halothane vapor with a vaporizer system. Mice were randomly divided into four groups, including the control group. Belinostat was dissolved in a mixture of 10% DMSO, 10% Cremophor EL, 80% water and a series of liposomal formulations (PC group, PS group, and PC/PS mixture group). It was then administered by subcutaneous injection with a dose of 0.33 mg/g. At time intervals of 5, 10 and 30 min and 1, 2, 3 and 4 h, 100 μL tail vein blood samples were collected in heparin-containing tubes (Vacutainer; BD, Franklin Lakes, NJ, USA) and immediately centrifuged at 3000× *g* for 10 min. The plasma was collected and stored at −20 °C. Mice were killed at the final time point and liver, spleen, lungs and kidneys were collected, washed with PBS and stored at −20 °C.

#### 4.12.3. Pretreatment of Plasma and Organ Samples

Plasma samples (40 μL) were mixed with 20 μL of internal standard solution (carbamazepine 20 μL/mL) and 1000 μL of methanol was added to induce protein precipitation. The mixture was centrifuged at 3000× *g* for 10 min and the supernatant was evaporated under vacuum for 2 h. The final dry residue was resuspended in methanol and analyzed by HPLC. Organs were homogenized in 1 mL PBS at 17,800 rpm for 30 s. Subsequently, following the same process, homogenized organs were mixed with internal standard solution and acetonitrile and analyzed by HPLC.

#### 4.12.4. Calculation of Pharmacokinetic Parameters

Correlation coefficients for the pharmacokinetic compartmental model were calculated and predicted using Phoenix WinNonlin V8.1 software (Certara, St. Louis, MO, USA). The area under the concentration–time curve (AUC_0–∞_) was calculated using the trapezoidal rule. Half-life (T_½_) and the clearance (Cl), volume of distribution (V_d_), absorption rate (Ka) and elimination rate (Ke) were also calculated by software.

### 4.13. Toxicity Evaluation

#### 4.13.1. Ex Vivo Hemolysis Assay

Red blood cells (RBCs) were donated by healthy volunteers. Collected blood was centrifuged at 2000 felative centrifugal force (RCF) for 10 min followed by redispersal three times in PBS (pH 7.4). Thereafter, the RBCs were dispersed in PBS (pH 7.4, 1:9 (v/v)) as the stock solution and stored at 4 °C for later use. To analyze hemolysis, 100 μL stock solutions were mixed with 50 μL belinostat free drug and belinostat-loaded liposomal formulations along with 2.45 mL PBS (pH 7.4). The mixture was incubated at 37 °C for 1 h and centrifuged at 2000 RCF for 5 min to separate the hemoglobin from cell debris. Finally, the supernatant was assayed at 415 nm using a UV spectrometer. The measured value for the positive control was RBCs dissolved in water and was taken as 100% hemolysis, while the value for the negative control was RBCs dissolved in PBS (pH 7.4) and was taken as 0% hemolysis.

#### 4.13.2. Peripheral Blood Mononuclear Cells (PBMCs)

Human peripheral blood mononuclear cells (PBMCs) were isolated from the peripheral blood from a healthy male volunteer. All procedures were performed with the approval of the institutional review board of Kaohsiung Medical University Hospital, Kaohsiung, Taiwan (KMUHIRB-E(I)-20190171). PBMCs were isolated by the density gradient centrifugation method by Ficoll-Paque™ PLUS (GE, Healthcare, Uppsala, Sweden). Afterwards, PBMC were collected from the interphase layer and washed three times with RPMI 1640 medium. PBMCs (1 × 10^6^ cells/well) were seeded in a 96 well plate and incubated at 37 °C, 5% CO_2_ in RPMI1640 with 10% FBS. After overnight incubation, the cytotoxicity assay was evaluated. The procedure was same as method 4.10.

#### 4.13.3. Biochemical Analysis

BALB/c mice were administered with subcutaneous injections with doses of 0.33 mg/g of belinostat free drug and belinostat-loaded liposomal formulations. Four hours after treatment, 100 μL tail vein blood samples were collected in heparin-containing tubes for complete blood count (CBC) analysis. Blood was analyzed for number of erythrocytes, platelets (Plt) and total white blood cells using an automated blood analyzer (XT-1800i; Sysmex, Kobe, Japan). Hemoglobin, red blood cells and hepatic function were also analyzed. Reference values were obtained from Charles River Laboratory. ALT and AST were also evaluated.

### 4.14. Statistical Analysis

Statistical analyses were performed using the *t*-test, one-way or two-way ANOVA and post hoc Tukey test by SigmaPlot 12.0. A 0.05 level of probability was taken as the level of significance. All data are expressed as the mean ± standard deviation (SD).

## 5. Conclusions

Controlled release pattern, improved burst effect and change to pharmacokinetics behavior are the key factors for improving the drug’s toxicity for use in chemotherapy. In summary, we developed a belinostat-loaded liposome system that showed superior characteristics and improved pharmacokinetics behavior. All prepared liposomes presented about 50 nm scale with excellent physical stability for 2 months. All prepared liposomes demonstrated sustained release behavior and the release mechanism was fitted with the Hixson–Crowell model. Based on the findings, we propose that phospholipid types are crucial factors for circulation behavior. The study demonstrated that using DOPC phospholipid material increased the half-life of belinostat, decreased clearance and presented a higher area under curve exposure. The DOPS group did not significantly improve pharmacokinetic behavior. Lymphatic delivery is complex. We will conduct a study localized to the lymphatic organs in the near future.

## Figures and Tables

**Figure 1 cancers-12-02558-f001:**
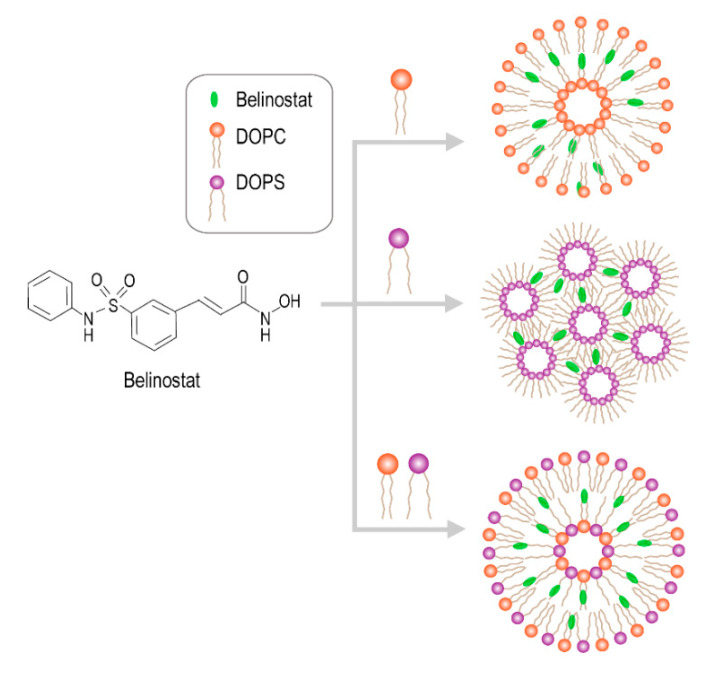
Schematic diagram of various liposomes incorporating dioleoylphosphatidylcholine (DOPC), dioleoylglycerophosphoserine (DOPS), and DOPC/DOPS mixture.

**Figure 2 cancers-12-02558-f002:**
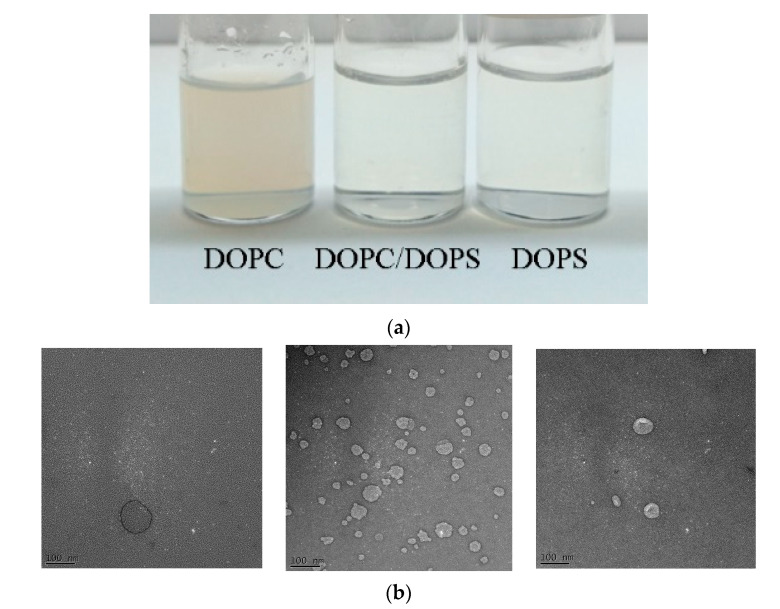
Optimization and characterization of the belinostat-loaded liposome delivery system. (**a**) Observation of the belinostat-loaded liposome delivery system. (**b**) Transmission electron microscopy (TEM) images of belinostat-loaded liposomes with DOPC, DOPS, or mixed DOPC/DOPS groups (left to right).

**Figure 3 cancers-12-02558-f003:**
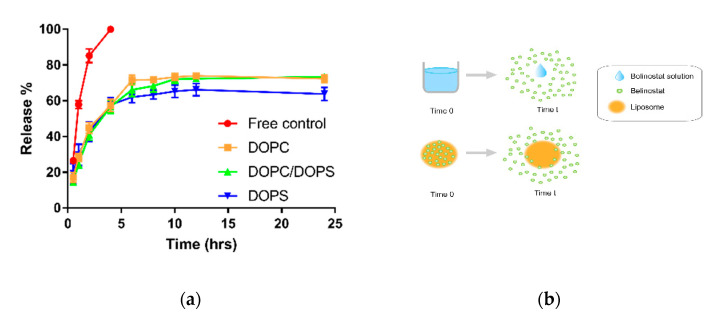
Drug release profiles and prediction by mathematical models to investigate the release mechanism. (**a**) In vitro release profile of free belinostat and belinostat-loaded liposomes obtained using the dialysis method. (**b**) Schematic diagram of belinostat release mechanism from the liposomal particulate system. Note: Belinostat was dissolved in absolute ethanol in the control group. Data are presented as mean ± standard deviation (*n* = 3).

**Figure 4 cancers-12-02558-f004:**
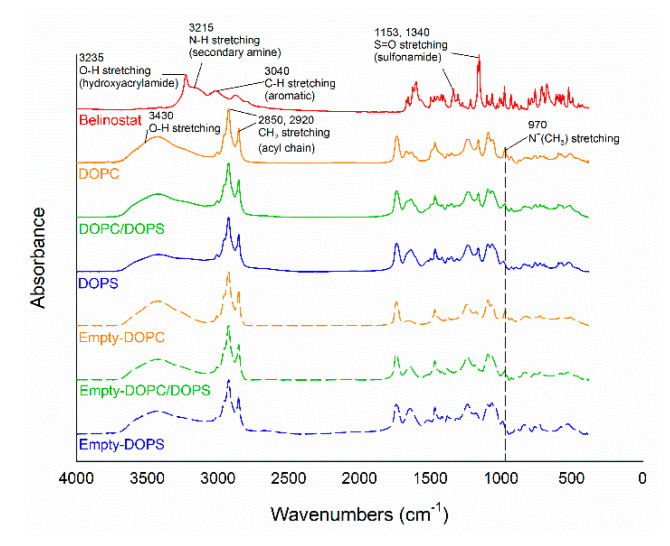
Fourier-transform infrared (FTIR) spectrum of belinostat free drug, empty liposomal formulations and belinostat-loaded liposomal formulations.

**Figure 5 cancers-12-02558-f005:**
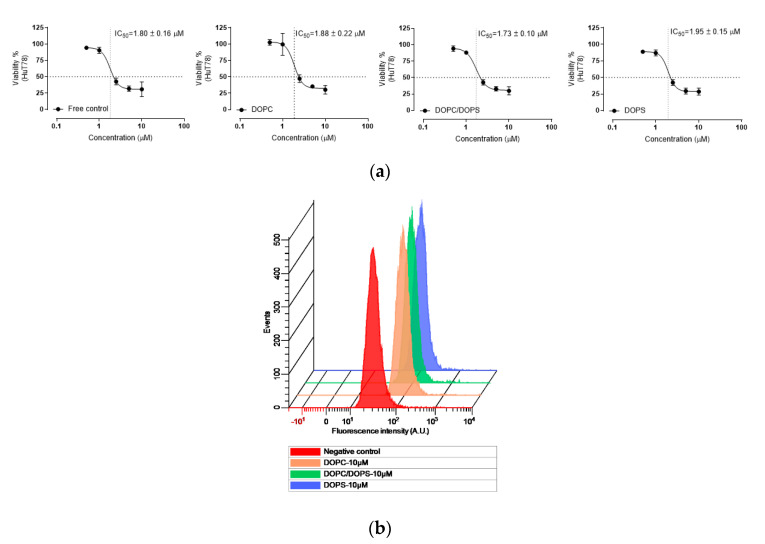
The cytotoxicity and cellular uptake study of belinostat-loaded liposomal formulations. (**a**) Dose response of belinostat free drug compared to belinostat-loaded liposomal formulations on HuT-78 cutaneous T-cell lymphoma cell line at the concentration of 0.5−10 μM after 72 h. (**b**) The qualitative analysis of flow cytometry measurement of HuT-78 treated with belinostat-loaded liposomal formulations at 10 μM for 4 h. Note: Rhodamine was used as a dye and was entrapped in different liposome formulations. Data are presented as mean ± standard deviation (*n* = 6). There were no significant differences.

**Figure 6 cancers-12-02558-f006:**
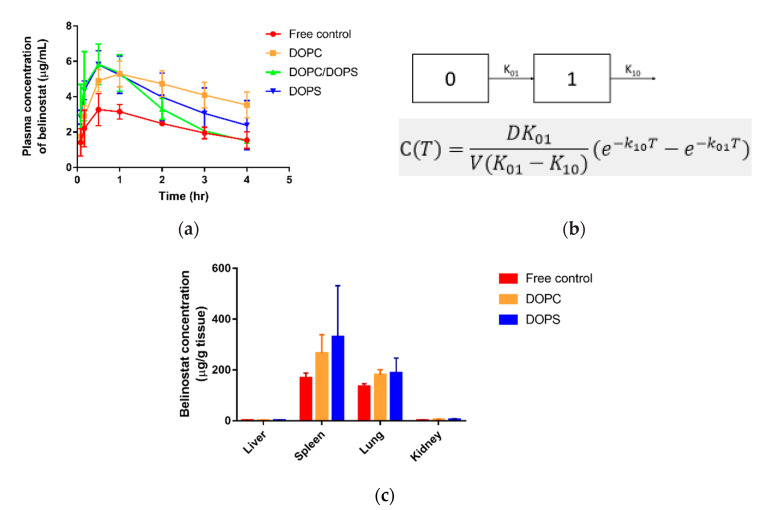
In vivo pharmacokinetics behavior. (**a**) Plasma concentration vs. time profile of belinostat free drug and belinostat-loaded liposomal groups with DOPC, DOPS, and DOPC/DOPS mixture groups after subcutaneous administration in Balb/c mice as a single dose of 0.333 mg/g. (**b**) The one-compartment pharmacokinetics model was fitted with first-order input and output, no lag time and estimated relative equation. (**c**) In vivo biodistribution of belinostat free drug and belinostat-loaded liposomal formulations. Data are presented as mean ± standard deviation (*n* = 3).

**Figure 7 cancers-12-02558-f007:**
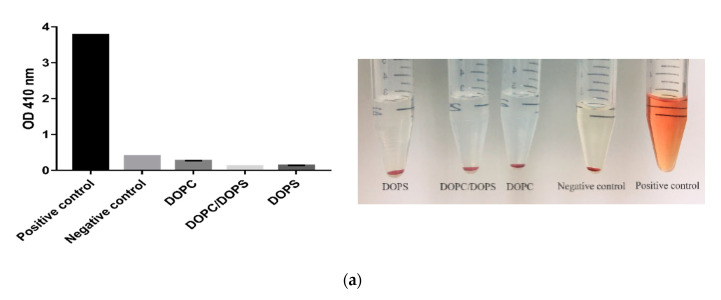
The safety assessment of liposomal formulations. (**a**) In vitro hemolysis of erythrocytes induced by liposomal formulations and observation. The positive control was RBCs dissolved in water and the negative control was RBCs dissolved in PBS (pH 7.4). (**b**) In vitro cytotoxicity of PBMCs. Notes: Data are shown as mean ± SD (*n* = 3). * *p* < 0.05compared to control group.

**Table 1 cancers-12-02558-t001:** Characterization of belinostat-loaded liposomes with DOPC, DOPS, and mixed PC/PS formulations in terms of particle size, polydispersity index (PDI), zeta potential, and entrapment efficiency.

Formulations	Particle Size (nm)	Polydispersity Index (PDI)	Zeta (mV)	Entrapment Efficiency (%)
DOPC	66.65 ± 3.15	0.116 ± 0.031	−10.18 ± 1.95 *	94.22 ± 2.20 *
DOPC/DOPS	52.62 ± 1.20 *	0.205 ± 0.006	−64.27 ± 4.01	65.08 ± 14.02
DOPS	54.28 ± 2.30 *	0.210 ± 0.012	−70.00 ± 5.79	59.75 ± 15.67

Note: Data are presented as mean ± standard deviation (*n* = 3). Significant differences are indicated with asterisks (* *p* < 0.05). Abbreviations: DOPC: 1,2-dioleoyl-sn-glycero-3-phosphocholine; DOPS: 1,2-dioleoyl-sn-glycero-3-phospho-L-serine.

**Table 2 cancers-12-02558-t002:** Comparison of suitable release pattern obtained from fitting experimental data with five models.

Release Models	Zero Order	First Order	Higuchi	Korsmeyer–Peppas	Hixson–Crowell
	Mt/M∞ = K0t	Mt/M∞ = 1−exp(−K1t)	$\mathrm{Mt}/M\infty =K\sqrt{t}$	Mt/M∞ = Ktn	$\sqrt[{3}]{M_{t}}-\sqrt[{3}]{M_{\infty }}\;$= Kt
Free	R2 = 0.911	R2 = 0.830	R2 = 0.979	R2 = 0.597	R2 = 0.057
DOPC	R2 = 0.905	R2 = 0.979	R2 = 0.993	R2 = 0.876	R2 = 0.994
DOPC/DOPS	R2 = 0.911	R2 = 0.975	R2 = 0.989	R2 = 0.894	R2 = 0.992
DOPS	R2 = 0.8421	R2 = 0.922	R2 = 0.982	R2 = 0.863	R2 = 0.996

Note: Data are presented as mean ± standard deviation (*n* = 3). Abbreviations: Mt/M∞: the fraction of drug released at time; K: release constant; R^2:^ coefficient of determination; n: swelling exponent.

**Table 3 cancers-12-02558-t003:** Comparative in vivo pharmacokinetic parameters of free drug and belinostat-loaded liposomal formulations.

Parameters	Control	DOPC Liposome	DOPS Liposome	DOPC/DOPS Liposome
AUC_0–∞_ (h * μg/mL	17.51 ± 5.99	43.99 ± 12.69	19.32 ± 6.03	26.86 ± 14.56
T_1/2_ (h)	3.29 ± 1.84	5.04 ± 1.62	1.79 ± 0.53	2.72 ± 1.27
Cl (L/h/kg)	404.50 ± 138.35	165.67 ± 65.25	365.28 ± 96.70	306.59 ± 169.65
Vd (L/kg)	1735.75 ± 414.86	1103.30 ± 145.32	905.60 ± 231.83 *	997.53 ± 100.99 *
T_max_ (h)	0.69 ± 0.31	0.87 ± 0.19	0.50 ± 0.23	0.51 ± 0.09
C_max_ (μg/mL)	3.39 ± 0.74	5.33 ± 0.71	6.05 ± 1.19 *	5.84 ± 0.79 *
K_a_ (1/h)	5.06 ±2.00	4.21 ±1.65	7.17 ±4.38	6.59 ±0.53
K_e_ (1/h)	0.25 ± 0.14	0.15 ± 0.06	0.42 ± 0.14	0.30 ± 0.17

Data are presented as mean ± standard deviation (*n* = 3). * *p* < 0.05 was considered statistically significant. Abbreviations: AUC: area under curve; T_1/2_: half-life; Cl: clearance; Vd: volume of distribution; T_max_: Time of maximum concentration observed; C_max_: maximum concentration observed; K_a_: absorption rate; K_e_: elimination rate.

**Table 4 cancers-12-02558-t004:** In vivo toxicity assessment of complete blood count and liver function assays of belinostat free drug and loaded liposome system in balbc mice after subcutaneous administration in balb/c mice at 0.333 mg/g dosage.

Items	Control	DOPC Liposome	DOPS Liposome	Reference Values
**Complete Blood Count**				
WBC count (10^9^/L)	3.42 ± 0.67	6.85 ± 0.78	5.70 ± 3.39	3.48–14.03
Lym (%)	78.21 ± 6.56	47.90 ± 3.68	45.21 ± 6.57	48.81–83.19
Mon (%)	10.95 ± 2.52	7.90 ± 0.14	3.45 ± 2.16	3.29–12.48
Gra (%)	13.47 ± 3.21	44.20 ± 3.54	38.32 ± 6.34	9.97–45.86
Hb (g/dL)	13.63 ± 2.16	17.45 ± 1.21	18.5 ± 1.84	12.6–20.5
Hct (%)	39.58 ± 3.80	57.65 ± 2.90	60.6 ± 6.51	42.1–68.3
RBC count (10^12^/L)	9.29 ± 1.28	11.40 ± 0.71	12.07 ± 1.23	6.93–12.24
Plt count (10^9^/L)	230 ± 45.72	529.5 ± 102.53 *	352.5 ± 50.20	420–1698
**Liver function markers**				
ALT (U/L)	125 ± 4.57	91.22 ± 3.53	48.08326 ± 16.26	41–131 (U/L)
ST (U/L)	290 ± 34.48	205.5 ± 44.5	155 ± 43.50	55–352 (U/L)

Notes: Data are shown as mean ± SD (*n* = 3). Abbreviations: WBC: white blood cell; Lym: lymphocyte; Mon: monocyte; Gra: granulocyte; Hb: hemoglobin; Hct: hematocrit; RBC: red blood cell; Plt: platelet. * *p* < 0.05 compared to control group.

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
