# Peer review of "Prolonging the Half-Life of Histone Deacetylase Inhibitor Belinostat via 50 nm Scale Liposomal Subcutaneous Delivery System for Peripheral T-Cell Lymphoma"

_cancers, 2020, doi:10.3390/cancers12092558_

Round 1

Reviewer 1 Report

The work by Cheng and collaborators, would like to demonstrate that the half-life of belinostat is improved by liposomal delivery. The manuscript is very difficult to read as there is no fluid consciousness. The results are just presented as they should stand alone and there is no explication on how they were obtained and no logical sequence of evidence. There is no evidence that the hypothesised prolonged half-life of the drug, has a real effect on HDAC.

Moreover, there are serious scientific flaws in the presented results.

In general is not belinostat with or without different liposomal formulation, but different liposomal formulation with or without belinostat, as the drug is loaded inside the liposomes and not viceversa.

Figure 2, panel a legend: where it came from the PE? In addition, what panel b should show? As for the legend, it should be “DOPC, DOPS, DOPC/DOPS liposome loaded with belinostat, instead of belinostat-loaded with…”. Panel c is totally unintelligible.

Figure 4. I could not understand how could be meaningful when the loaded liposomes do not show any firm of the loaded drug.

Figure 5. Panel a. None of the dose response graphs intercept the x-axis values at 50, so how could it be that these values. Are IC50?!?

Panel b. What should this panel mean? Better to have some values on the significance of fluorescence intensity on multiple samples, than a single histogram, which is not indicative of the cellular uptake.

Figure 6, panel d. None of the values have statistical significance, so how could the authors state it?

Figure 7, panel a. It is not clear what the reader should understand, as there is no explication in the text. Which is the positive control? Which is the negative control?

Panel b. I really could not understand this panel. There is no control, and the statistical significance seems to be artefactual.

Reviewer 2 Report

Dear Editor,

I reviewed the mauscript by Cheng et al., entitled “  Prolonging the half-life of histone deacetylase inhibitor belinostat via 50 nm scale liposomal subcutaneous delivery system for peripheral T-cell lymphoma “

The manuscript is interesting and may be well  designed and written. However, this study needs additional experiments to confirm the clinical validity of the discussed  techniques of the present manuscript. For example how should function the liposomes to deliver the anticancer agents of interist . At least the mechanisms whereby  liposomes deliver specifically the anticancer agents. This may improve the quality of the present study.

many thanks

Reviewer 3 Report

Authors describe the pharmacokinetics of a liposomal formulation for belinostat. The manuscript is well written and clear. Here some minor comments. 

  • Authors should also mention the previous work carried out with liposomes and the advantages of their platform in the introduction (https://doi.org/10.1016/j.ijpharm.2010.06.046)
  • Were the differences in Fig 6D statistically significant?
  • The authors should also include in the discussion if the pharmacokinetic results achieved in this manuscript are compatible with the drug administration in liposomes.
  • Fount type in the introduction.

Round 2

Reviewer 2 Report

Dear Editor,

reviewed the manuscript by Cheng et al., entitled” Prolonging the half-life of histone deacetylase

inhibitor belinostat via 50 nm scale liposomal subcutaneous delivery system for peripheral T-cell lymphoma    “

The manuscript is now improved and can be published in the present form.

Many thanks